# Combination of Kaempferol and Docetaxel Induces Autophagy in Prostate Cancer Cells In Vitro and In Vivo

**DOI:** 10.3390/ijms241914519

**Published:** 2023-09-25

**Authors:** Qian Zhou, Gang Fang, Yuzhou Pang, Xueni Wang

**Affiliations:** 1Guangxi Zhuang Yao Medicine Center of Engineering and Technology, Guangxi University of Chinese Medicine, Nanning 530200, China; 2Guangxi Key Laboratory of Applied Fundamental Research of Zhuang Medicine, Guangxi University of Chinese Medicine, Nanning 530001, China; 3Guangxi Higher Education Key Laboratory for the Research of Du-Related Diseases in Zhuang Medicine, Guangxi University of Chinese Medicine, Nanning 530001, China

**Keywords:** kaempferol, docetaxel, prostate cancer, combined treatment, autophagy

## Abstract

Docetaxel is a first-line chemotherapy drug used to treat advanced prostate cancer, but patients who have used it often face the challenges of drug resistance and side effects. Kaempferol is a naturally occurring flavonol; our previous studies have confirmed that it has excellent anti-prostate activity. To investigate the anti-prostate cancer effects of docetaxel in combination with kaempferol, we conducted experiments at the cellular and whole-animal level. Plate cloning assays showed that the combination of docetaxel and kaempferol had a synergistic effect in inhibiting the proliferation of prostate cancer cells. The combination of these two compounds was found to induce autophagy in prostate cancer cells via transmission electron microscopy, and changes in the expression of autophagy-related proteins via Western blot assays also confirmed the occurrence of autophagy at the molecular level. We also confirmed the anti-prostate cancer effect of docetaxel in combination with kaempferol in vivo by establishing a mouse xenograft prostate cancer model. Autophagy-related proteins were also examined in mouse tumor tissues and verified the presence of autophagy in mouse tumor tissues. The above cellular and animal data suggest that docetaxel in combination with kaempferol has significant anti-prostate cancer effects and that it works by inducing autophagy in cells.

## 1. Introduction

Prostate cancer is the second most common type of cancer worldwide and the fifth leading cause of cancer deaths [1].There are a variety of treatment strategies available to treat prostate cancer, including active surveillance, surgery, radiotherapy, chemotherapy, hormone therapy and immunotherapy [2]. Initially, these treatments prolonged patients’ lives and improved their quality of life. Eventually, however, these treatments fail one by one due to the progression of the cancer, the development of drug resistance and serious side effects. In this case, doctors need to update their patients’ treatment regimens, and these new regimens rely on scientists to provide new technologies and new medicines. Approximately 60% of the anti-cancer drugs approved by the FDA are made from natural ingredients [3]. Natural products have long been an important source for the discovery of new medicines.

Docetaxel is a derivative of the natural product paclitaxel, which has a broad-spectrum anti-cancer effect similar to paclitaxel. Docetaxel works by altering the microtubule dynamics of cells, which leads to cell cycle arrest during mitosis [4,5]. It was approved in 2004 for the treatment of men with metastatic castration-resistant prostate cancer and is now the standard of care for the disease [6]. Recent clinical trials have shown that patients with metastatic castration-sensitive disease and those who may have high-risk localized prostate cancer also benefit from docetaxel administration, expanding the role of chemotherapy in the treatment of prostate cancer [7]. However, the non-specific targeting of docetaxel, multi-drug resistance and multiple side effects are major barriers to the continued treatment of prostate cancer with docetaxel [8].

In order to overcome the problems arising from the treatment of prostate cancer with docetaxel, many scientists have opted for a combination strategy, evaluating the effects of docetaxel in combination with other drugs. For example, docetaxel has been used in combination with a variety of drugs, including tyrosine kinase inhibitors, antiangiogenic drugs, bone-targeting agents, Bcl-2 inhibitors, chemotherapy, immunological agents and vitamin D analogs [9]. Currently, some cases treated with docetaxel in combination with other drugs have reported positive changes in prostate-specific antigen levels, tumor size and survival [10,11]. There have also been cases where the combination of drugs with docetaxel has caused additional side effects in patients, such as toxicity and thromboembolism [12,13]. There have also been trials combining docetaxel with natural products to treat prostate cancer. As expected, the combination of natural products with docetaxel has been shown to have synergistic effects in the treatment of prostate cancer [14,15].

Kaempferol is a flavonol widely found in various herbs, vegetables and fruits. In our previous study, we found that it modulates androgen receptor signaling and has anti-benign prostatic hyperplasia effects [16]. We also found that it was able to inhibit both androgen-dependent and androgen-independent prostate cancer by modulating Ki67 expression [17]. Based on these studies, we have confirmed the anti-prostate cancer effects of kaempferol. However, it is not yet known what anti-prostate cancer effect can be achieved when kaempferol is combined with docetaxel. In this study, we will investigate the anti-prostate cancer effects of kaempferol combined with docetaxel at both the cellular and animal level. It may provide some clues for the development of new therapies for prostate cancer.

## 2. Results

### 2.1. Kaempferol Combined with Docetaxel Inhibited Prostate Cancer Cell Proliferation

We observed the effects of kaempferol combined with docetaxel on the proliferation of human prostate cancer PC-3 cells and DU145 cells via a plate clone formation assay. We found that docetaxel at 0.1–10 nM inhibited the proliferation of PC-3 cells in a dose-dependent manner; 10 μM of kaempferol also effectively inhibited the proliferation of PC-3 cells (Figure 1A,B). When 10 μM of kaempferol was used in combination with different concentrations of docetaxel, it was found that the combined effect of both of them was better than that of docetaxel alone (Figure 1A,B). Also, we obtained similar results on DU145 cells (Figure 1C,D). These data suggest that the combination of kaempferol and docetaxel may have a synergistic effect.

### 2.2. Kaempferol Combined with Docetaxel Triggers Autophagy in Prostate Cancer Cells

To understand the effect of the combination of the two drugs on the survival status of prostate cancer cells, we collected cells for transmission electron microscopy observation 72 h after PC-3 cells received different treatments. It is well known that autophagy is characterized by the presence of vesicles with a double-membrane structure, called autophagosomes, which contain cytoplasmic components, which subsequently fuse with lysosomes to form autolysosomes for degradation [18]. We found that after 72 h of compound treatment, autophagosomes (Figure 2A, Appendix A) as well as autolysosomes (Figure 3, Appendix A) were present in all three groups except the control group. Therefore, we inferred that cellular autophagy occurred in the kaempferol group, the docetaxel group, and the two-drug combination group after drug administration. To further confirm that autophagy occurred in PC-3 cells, we examined the levels of cellular autophagy-related proteins via a Western blot assay.

LC3 (microtubule-associated protein light chain 3) is a specific autophagy marker in mammalian cells and is involved in the formation of autophagosomes as well as the recruitment of autophagic cargo [19]. LC3-I is lipidated to LC3-II and bound to cargo isolation membranes during the induction of autophagic degradation, resulting in the formation of autophagosomes. The lipidation of LC3 leads to an increased LC3-II/LC3-I ratio which serves as a frequently used marker of autophagy [20]. Currently, there is a general consensus that p62 plays a role as both an effector of selective autophagy and a substrate to autophagy [21]. Through its LIR domain, the p62 protein interacts with LC3 for attachment to the autophagosomes, subsequently delivering the ubiquitinated cytotoxic target materials attached to it through its UBA domain for transportation to the lysosome and degradation [22,23]. In this study, we investigated the expression of LC3 and p62 in PC-3 cells following treatment with the control group, the kaempferol group, the docetaxel group, and the two-drug combination groups. A significant difference was observed between the group treated with kaempferol combined with docetaxel and the control group. Additionally, the combination of these two drugs led to a notable increase in the expression of LC3-II protein (*p* < 0.01) (Figure 2B and Figure 4A,B) and p62 protein (*p* < 0.05) (Figure 2C and Figure 4C,D). Furthermore, there was an increase in the LC3-II/LC3-I ratio by calculation (*p* < 0.01) (Figure 2B). These results of autophagy-specific proteins suggest that the combination of the two drugs increased autophagy in PC-3 cells. To validate this result, the experiment was performed more than three times to confirm the accuracy and generalizability of the finding. Overall, the results indicate that the combination of kaempferol and docetaxel can induce cellular autophagy in PC-3 prostate cancer cells.

### 2.3. Kaempferol Combined with Docetaxel Inhibits Prostate Cancer Tumor Growth and Triggers Autophagy in Prostate Cancer Tumor In Vivo

To investigate whether the combination of these two compounds could work in vivo, we generated an in vivo prostate cancer model by subcutaneously injecting PC-3 cells into nude mice to study the effect of kaempferol and docetaxel combination on prostate cancer in vivo. Once the subcutaneous tumor volume of nude mice reached 100 cubic millimeters, the nude mice were grouped and drug administration commenced. Tumor volume and body weight were monitored twice a week and duly noted. The tumor volume data depicted that after 28 days of administration, the group administered with 10 mg/kg kaempferol along with 1 mg/kg docetaxel showed a significantly smaller tumor volume than the control group (*p* < 0.05) (Figure 5A,B). However, there was no significant difference noted in the tumor volume between the group administered with 1 mg/kg docetaxel alone and the control group (*p* > 0.05). We analyzed the mice’s body weight changes during drug administration. No notable difference was observed in the body weights of the nude mice across the dosing groups after 28 days of treatment (Figure 5D). The combination of kaempferol and docetaxel notably led to a substantial decrease in tumor size and weight in nude mice after 28 days of administration compared to the control group (*p* < 0.05) (Figure 5A,C). In contrast, no significant change in tumor volume was observed in the group treated with docetaxel alone, compared to the control group (*p* > 0.05) (Figure 5A,C). This evidence demonstrated that the therapeutic effect of kaempferol combined with docetaxel was superior to that of the group treated with docetaxel alone. Kaempferol and docetaxel combination effectively inhibited the development of prostate cancer in nude mice.

To confirm whether autophagy induced by the combination of kaempferol and docetaxel in prostate cancer cells also occurs in nude mice, we examined changes in the expression of the autophagy signature proteins LC3 protein and beclin1 protein in the tumor tissue of the aforementioned groups. The Western blotting analysis demonstrated decreased expression of LC3-I and increased expression of LC3-II in the treatment group that received the combination of kaempferol and docetaxel compared to the control group (Figure 6A). Calculation also revealed a higher LC3-II/LC3-I ratio in the group treated with the combination of the two drugs compared to the control group. Moreover, the expression of the beclin1 protein was higher in the group treated with the combination of the two drugs compared to the control group (Figure 6B). As the substrate of autophagy, the expression of p62 is at a dynamic level during the occurrence of autophagy [24]. In the cells, we detected the expression of p62 after 48 h of compound treatment. At this time, the cells were undergoing autophagy, accumulating a large amount of p62 as the substrate for autophagy. Therefore, the p62 expression we detected increased. Contrary to the cellular level results, we detected a decrease in p62 expression in mouse prostate cancer tissue (Figure 6C). This may be because after one month of continuous administration, the autophagy of cells in mice enters the late stage, so the detected p62 is reduced. Based on these data, it can be concluded that the combination of kaempferol and docetaxel induces autophagy both in vitro and in vivo. These results confirm that the combination of kaempferol and docetaxel can induce autophagy in prostate cancer in vivo.

According to our previous study, kaempferol induces apoptosis in PC-3 cells. However, it was unclear whether the combination of kaempferol and docetaxel could also induce PC-3 apoptosis. Subsequently, we investigated the expression of Caspase-3 and Beta Catenin proteins in the tumor tissues of the aforementioned groups. Caspase-3 is the most downstream enzyme in the signaling pathway that triggers apoptosis, and its cleavage indicates that the cell undergoes apoptosis [25]. Beta Catenin regulates cell-to-cell adhesion, cell proliferation, and a range of physiological activities, including cell differentiation and apoptosis [26,27]. According to current evidence, the upregulation of Beta Catenin hinders apoptosis in PC-3 cells [28]. Protein blotting analysis did not reveal any notable alteration in the quantity of protein for the full length of caspase 3 in the tumor tissue of mice that were administered the two-drug combination, as compared to the control group (Appendix A). The expression of Beta Catenin protein increased in the group that received the two-drug combination compared to the control group (Figure 6C). These findings confirm that the combination of kaempferol and docetaxel did not stimulate cell apoptosis in prostate cancer cells. 

Research indicates that epithelial-to-mesenchymal transition (EMT) plays a role in the metastatic and invasive processes of prostate cancer [29]. E-cadherin is involved in intercellular adhesion, epithelial-to-mesenchymal transition and the migration and invasion of cancer cells [30]. Cell migration and invasion in prostate cancer can be significantly inhibited through E-cadherin overexpression [31,32]. Neuro(N)-cadherin is a single-chain transmembrane glycoprotein that is dependent on calcium and mediates homotypic and heterotypic cell–cell adhesion [33]. The reduced expression of N-cadherin leads to an impact on prostate cancer cell invasion, migration and epithelial-to-mesenchymal transition [34,35]. In our experiment, the two-drug combination group increased the expression of E-cadherin protein (Figure 6E) and decreased the expression of N-cadherin protein in the tumor tissues of the aforementioned groups (Figure 6F) compared with the control group. This result suggests that kaempferol combined with docetaxel can inhibit prostate cancer tumor migration.

## 3. Discussion

Currently, there are multiple docetaxel combinations available. In the E3805 (CHAARTED) study, the combination of docetaxel and androgen deprivation therapy was found to significantly enhance overall survival in patients with metastatic hormone-sensitive prostate cancer [36]. Furthermore, docetaxel can be synergistically combined with other biologics. The co-treatment of docetaxel and FAK inhibitor VS-6063 resulted in a significantly greater reduction in the viability of docetaxel-resistant CRPC cells and the stronger inhibition of PC-3 xenograft growth when compared to monotherapy alone [37]. The combination of docetaxel with anti-PD1 blockade yielded better prostate-specific antigen progression-free survival compared to anti-PD1 blockade alone [38]. Of note, docetaxel can also be combined with natural products. For instance, combining Aneustat with docetaxel has shown a significant and synergistic increase in anti-prostate cancer activity both in vitro and in vivo. Moreover, in vivo experiments have demonstrated that the combination does not cause significant host toxicity [39]. Due to their market availability, low cost, low toxicity, potential for low pharmacological interactions and possible synergism with anti-cancer drugs, natural products are expected to be the next major drug combined with docetaxel.

Compounds usually work by killing cancer cells, and a variety of cell death modes have been identified, including necrosis, pyroptosis, apoptosis, necroptosis, ferroptosis, cuproptosis and autophagy [40]. Autophagy is a process by which the cell engulfs its own cytoplasmic proteins or organelles, encapsulates them in vesicles and fuses them with lysosomes to form autophagic lysosomes, which degrade their encapsulated contents to meet the metabolic needs of the cell itself and to renew certain organelles [41]. In prostate cancer, autophagy plays a key role in regulating cell survival. Depending on the environment of the cell, autophagy can be either negative or positive for the development of prostate cancer [42]. It has been found that compounds that induce autophagy in prostate cancer cells can be used to kill the cells. For example, diosgenin, a naturally occurring steroid compound isolated from *Dioscorea* spp., significantly reduces the proliferation and viability of prostate cancer cells by inducing autophagy [43].

Our current experiment demonstrated that the combination of docetaxel and kaempferol was more effective in terms of efficacy than either drug used alone. We used a plate cloning assay to make this determination. We also observed, by means of transmission electron microscopy, the production of autophagic vesicles in PC-3 prostate cancer cells after drug administration. Furthermore, we confirmed the changes in autophagy-related proteins by extracting PC-3 cell proteins after drug administration. Our results demonstrated a significant increase in the protein expression of LC3-II and p62 when the two drugs were used in combination. It was concluded that the combination of two drugs increased autophagy in PC-3 cells. Furthermore, the combination of the two drugs was found to effectively inhibit prostate tumor growth when studied using an animal xenograft tumor model. The tumor inhibition effect of the combination of the two drugs is superior to that of a single drug. Finally, we extracted the tumor proteins and observed similar expression of autophagy-related proteins in the tumor compared to that at the cellular level. Thus, our in vitro and in vivo experiments have confirmed that the combined treatment of the two drugs is more effective for treating prostate cancer compared to using any single drug.

The role of autophagy in tumor development is intricate. It can promote or inhibit tumor growth at different stages or under different microenvironmental conditions [44]. The combination of two drugs induced autophagy in PC-3 prostate cancer cells, which inhibited prostate cancer development in our experiments. In contrast, when acting alone, docetaxel kills cells by inducing apoptosis. This finding suggests that adding kaempferol to continue treatment may be a strategy to overcome resistance when cancer cells develop resistance to docetaxel treatment. The combination of two drugs, kaempferol and docetaxel, is concluded in this study to shift the induction of apoptosis into autophagy induction as an anti-prostate cancer approach. This provides a new therapeutic strategy for the development of therapies for prostate cancer.

## 4. Materials and Methods

### 4.1. Reagents and Antibodies

Kaempferol (S2314) was purchased from Selleck (Shanghai, China). DMEM/F-12 (1:1) basic (1×) (C11330500BT) and MEM Alpha basic (1×) (C12571500CP) were purchased from Gibco (Jiangsu, China). Fetal bovine serum (FBSSR-01021-500) was purchased from OriCell bioproducts (Shanghai, China). Trypsin–EDTA solution (T1300), DMSO (D8371), Penicillin–Streptomycin solution (P1400), RIPA buffer (R0020), PMSF (P0100), Bovine Serum Albumin V (A8020), PBS, 1× (pH 7.2–7.4, 0.01M, cell culture) (P1020), Glutaraldehyde, 2.5% (EM Grade) (P1126) and Hoechst 33342 Stain solution (C0031) were purchased from Solarbio (Beijing, China). A Pierce™ BCA Protein Assay Kit (23227), NuPAGE™ LDS Sample Buffer (4×) (NP0007), NuPAGE™ Sample Reducing Agent (10×) (NP0009), NuPAGE™ MES SDS Running Buffer (20×) (NP0002), Power Blotter1-Step™ Transfer Buffer (5×) (PB7300) and NuPAG™ 10% Bis-Tris Gel (NP0302BOX) were purchased from Thermo Fisher Scientific (Waltham, MA, USA). Immobilon^®^-PSQ Transfer Membrane (ISEQ00010) and Immobilon^®^ Western Chemiluminescent HRP Substrate (WBKLS0100) was purchased from Merck Millipore Ltd. (Burlington, MA, USA) All antibodies were purchased from proteintech (Rosemont, IL, USA), including LC3 Recombinant antibody (81004-1-RR), p62, SQSTM1 Polyclonal antibody (18420-1-AP), Beclin 1 Monoclonal antibody (66665-1-Ig), Caspase 3/p17/p19 Polyclonal antibody (19677-1-AP), E-cadherin Polyclonal antibody (20874-1-AP), N-cadherin Monoclonal antibody (66219-1-Ig), Beta Catenin Mouse Monoclonal antibody (66379-1-Ig) and Vinculin Polyclonal antibody (26520-1-AP).

### 4.2. Cell Culture

PC-3 cells and DU145 cells were obtained from the Committee of Type Culture Collection of the Chinese Academy of Sciences (Shanghai, China). PC-3 cells were cultured in DMEM/F12 supplemented with 10% (*v*/*v*) fetal bovine serum (FBS), 100 units/mL penicillin and 100 μg/mL streptomycin. DU145 cells were cultured in MEM supplemented with 10% (*v*/*v*) fetal bovine serum (FBS), 100 units/mL penicillin and 100 μg/mL streptomycin.

### 4.3. Plate Clone Formation Assay

PC-3 cells and DU145 cells were inoculated into 6-well plates and incubated with 10% FBS DMEM/F12 and 10% FBS MEM for 48 h. PC-3 cells and DU145 cells were then treated with fresh medium containing different concentrations of the compounds for one week. After one week, the cells were treated again with the same fresh medium for another week. The cells were then fixed with 4% paraformaldehyde for 15 min and stained with 0.1% crystal violet solution for 30 min. The cells were then washed with PBS buffer and allowed to evaporate at room temperature. Finally, cell colonies were imaged and counted using a colony-forming cell analysis system (Gelcount, Oxford Optronix, Adderbury, UK).

### 4.4. Western Blot

PC-3 cells were inoculated into 10 cm cell culture dishes and incubated with 10% FBS DMEM/F12 for 48 h. PC-3 cells were then treated with different concentrations of compounds for 72 h. After 72 h, compound-treated PC-3 cells were collected, and then, the cells were lysed with a mixture of RIPA lysis buffer and PMSF for 30 min at 4 °C, during which time, the solution was mixed several times. Tumor tissue from nude mice, approximately 50 mg of each sample, was first cut up with scissors; then, the tumor tissue was lysed in a mixture of RIPA lysis buffer and PMSF for 30 min at 4 °C, during which time, the tissues were lysed with the aid of an ultrasonic crusher. The lysates were collected via centrifugation, and the total protein content was determined using a BCA protein assay kit (23227, Thermo Fisher Scientific). The prepared total protein samples were separated via SDS-PAGE gel electrophoresis. Proteins were then transferred onto 0.2 µm polyvinylidene difluoride (PVDF) membranes (ISEQ00010, Merck Millipore Ltd.) using a Power Blotter Station (PB0010, Thermo Fisher Scientific). The protein-loaded PVDF membrane was then blocked with 5% skimmed milk for 1 h and incubated with the primary antibody for 18 h at 4 °C. At the end of the incubation, the membrane was washed with TBST buffer. The PVDF membranes were then immersed in secondary antibody solution for 1 h at room temperature and washed with TBST buffer at the end of the reaction. Finally, the protein blots were detected with the chemiluminescent substrate and photographed using a Chemidoc CD Touch (Bio-Rad, Hercules, CA, USA). Finally, the protein blots were analyzed using Image lab 6.0 software (Bio-Rad, Chinese edition).

### 4.5. Transmission Electron Microscopy

PC-3 cells were inoculated into 10 cm cell culture dishes and incubated with 10% FBS DMEM/F12 for 48 h. PC-3 cells were treated with DMSO (0.1%, *v*/*v*), docetaxel (10 nM), Kae (20 μM), docetaxel (10 nM) vs. Kae (20 μM) for 72 h. After 72 h, 3 mL of Glutaraldehyde, 2.5% (EM Grade) (P1126, Solarbio) was added to each Petri dish and left for 10 min. The cells were scraped up with a cell scraper (179693, Thermo Fisher Scientific) and transferred to a 15 mL centrifuge tube. Cells containing 2.5% glutaraldehyde fixative were centrifuged at 1500 rpm for 5 min. At the end of centrifugation, the supernatant was discarded. Cells were re-fixed with fresh Glutaraldehyde, 2.5% (EM Grade) for 48 h. The cells were centrifuged and the supernatant was discarded, and then the cells were washed by adding 0.1 M phosphate buffer PB (pH 7.4). The cells were centrifuged and the supernatant was discarded. The cells were wrapped in a 1% agarose solution. Cells wrapped in 1% agarose solution were fixed in 1% OsO_4_ (18456, Ted Pella Inc., Redding, CA, USA) for 2 h at room temperature and protected from light. The cells were then rinsed 3 times with 0.1 M phosphate buffer PB (pH 7.4). Cells were dehydrated with gradient ethanol (100092183, Sinaopharm Group Chemical Reagent Co. Ltd., Shanghai, China). The cells were then dehydrated twice with 100% acetone (10000418, Sinaopharm Group Chemical Reagent Co., Ltd.). The pure EMBed 812 (90529-77-4, SPI, Rochester, NY, USA) was poured into the embedding models, and the cell was inserted into the pure EMBed 812, and then, it was kept at 37 °C overnight. The embedded models were polymerized in an oven at 60 °C for 48 h, and the resin blocks were removed and set aside. The resin blocks were ultrasonically sectioned using an ultramicrotome (Leica UC7, Leica, Wetzlar, Germany) at 60–80 nm. The cuprum grids were observed under a transmission electron microscope (HT7800, hitachi, Tokyo, Japan), and images were taken.

### 4.6. Immunofluorescence Staining

Cellular immunostaining was performed according to the method of our previous study [17]. Briefly, PC-3 cells were cultured on 96-well assay plates (black plates) pretreated with 10% L-polylysine. After treatment with the compounds, the cells were returned to the incubator for a further 48 h. The cells were then subjected to a series of treatments including fixation, permeabilization, washing and containment. The cells were then incubated with LC3 or p62 antibodies overnight at 4 °C, followed by incubation with CoraLite488-conjugated Affnipure sheep anti-rabbit IgG (H + L) for 1 h at room temperature. The nuclei were then stained with Hoechst 33342 for 20 min and the cells were rinsed with PBS. The cells were analyzed using a Perkin Elmer High Content Screening system (Operetta, PerkinElmer) to obtain images at 20× magnification.

### 4.7. The Xenograft Tumor Model

This animal experiment was approved by the Guangxi University of Chinese Medicine Institutional Animal Welfare and Ethical Committee (approval code: No. DW20220926-199) for studies involving animals. Fifteen male Balb/c mice (age = 5–6 weeks) were obtained from Hunan SJA laboratory animal Co., Ltd., Changsha, China. All Balb/c mice were housed in an SPF room and received 12 h of light and 12 h of darkness daily, with free access to sterile food and water. After 7 days of adaptation, each mouse was subcutaneously inoculated with 5 × 10^5^ androgen-independent PC-3 human prostate cancer cells. The intervention started when tumors reached a volume of 50–100 mm^3^ about one week after inoculation. Fifteen BALB/nude mice were randomly divided into three groups (n = 5): the control group, 1 mg/kg docetaxel group, and 1 mg/kg docetaxel combined with 10 mg/Kg kaempferol group. Docetaxel and kaempferol were dissolved in a solvent containing 5% DMAO, 30% PEG 300, 5% Tween 80 and 60% H_2_O. Throughout the study, docetaxel was administered intraperitoneally twice a week for four weeks in the 1 mg/kg docetaxel group. The 1 mg/kg Docetaxel combined with 10 mg/Kg kaempferol group were given daily doses of kaempferol and were injected intraperitoneally with docetaxel twice a week for 4 weeks. Tumor size was measured twice a week with calipers. Tumor volume was calculated by using the following formula: π/6 × ength × width × width [45]. Mouse body weight was measured twice a week. Mice were sacrificed when they had received a 4-week intervention treatment in observance of the institutional guideline on tumor size. Xenograft tumors were harvested, weighed and fixed with 4% paraformaldehyde and/or snap frozen and stored in liquid nitrogen.

### 4.8. Statistical Analysis

All results were presented as mean ± standard deviation (SD). Statistical significance was determined with One-Way ANOVA. * *p* < 0.05, ** *p* < 0.01 was considered statistically significant.

## Figures and Tables

**Figure 1 ijms-24-14519-f001:**
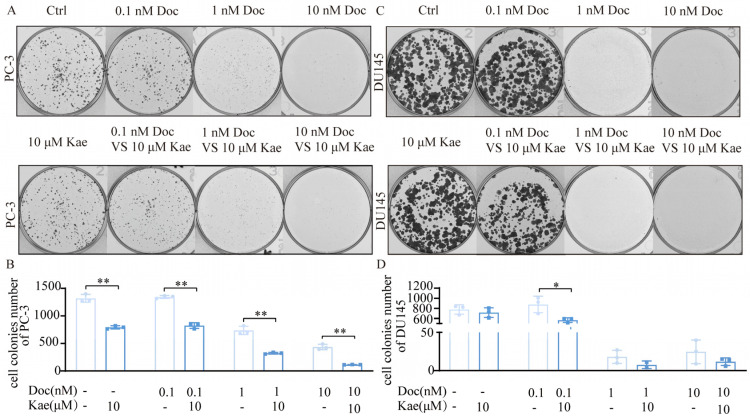
Effect of kaempferol combined with docetaxel on the proliferation of prostate cancer cells. (**A**) PC-3 cells exposed to different treatments for 14 days; (**B**) number of clones of PC-3 cells after different treatments; (**C**) DU145 cells exposed to different treatments for 14 days; (**D**) number of clones of DU145 cells after different treatments. * *p* < 0.5, ** *p* < 0.01.

**Figure 2 ijms-24-14519-f002:**
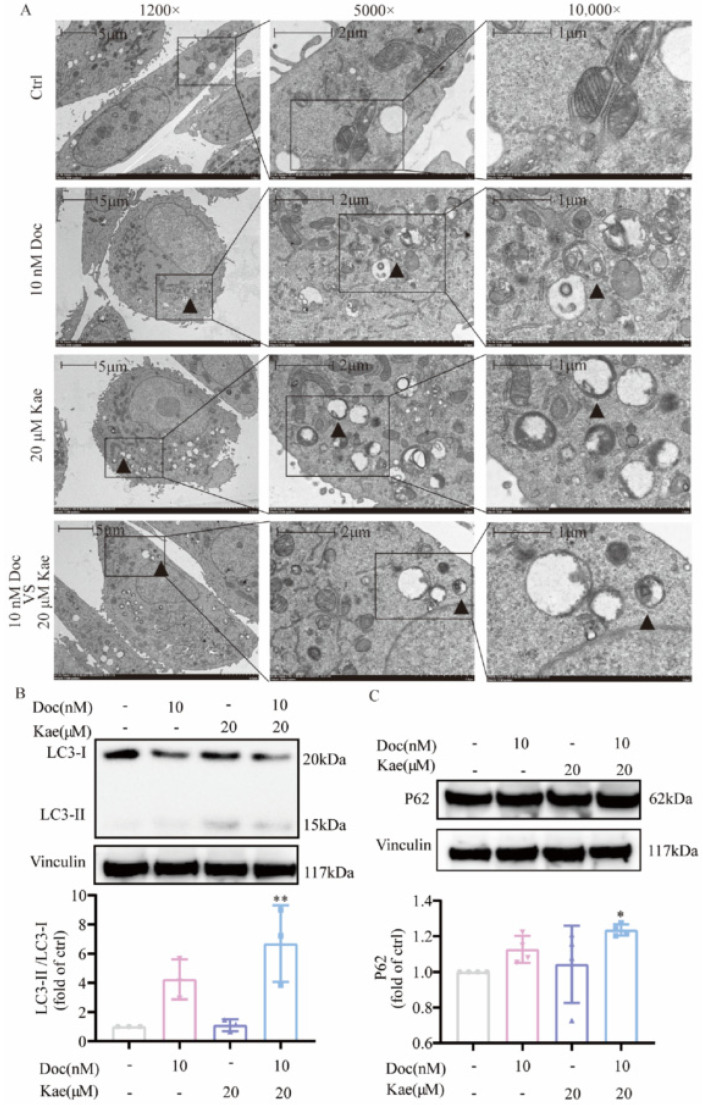
Kaempferol and kaempferol in combination with docetaxel induced autophagy in PC-3 cells. (**A**) Observation of autophagy in PC-3 cells using transmission electron microscopy; (**B**) changes in the expression of autophagy-related proteins. * *p* < 0.5, ** *p* < 0.01 vs. Ctrl.

**Figure 3 ijms-24-14519-f003:**
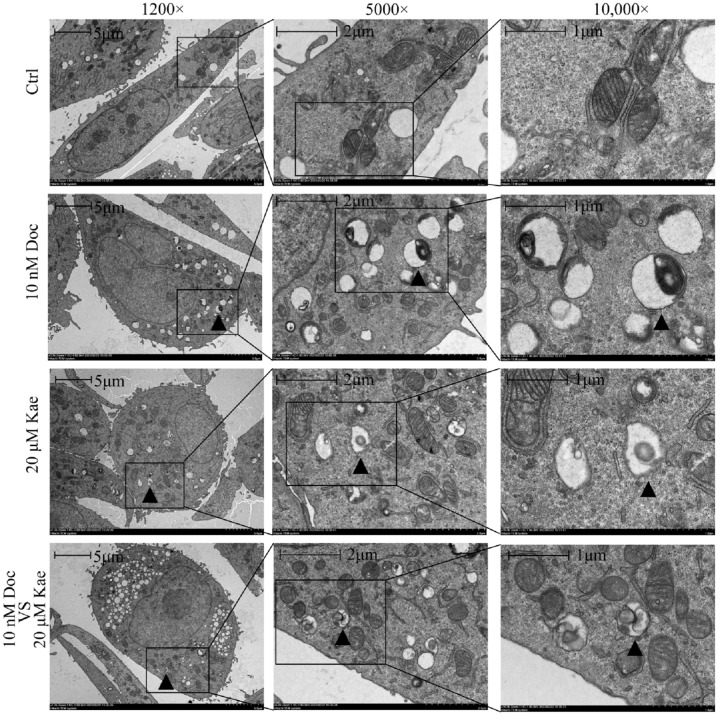
Examining autolysosomes in PC-3 cells via transmission electron microscopy.

**Figure 4 ijms-24-14519-f004:**
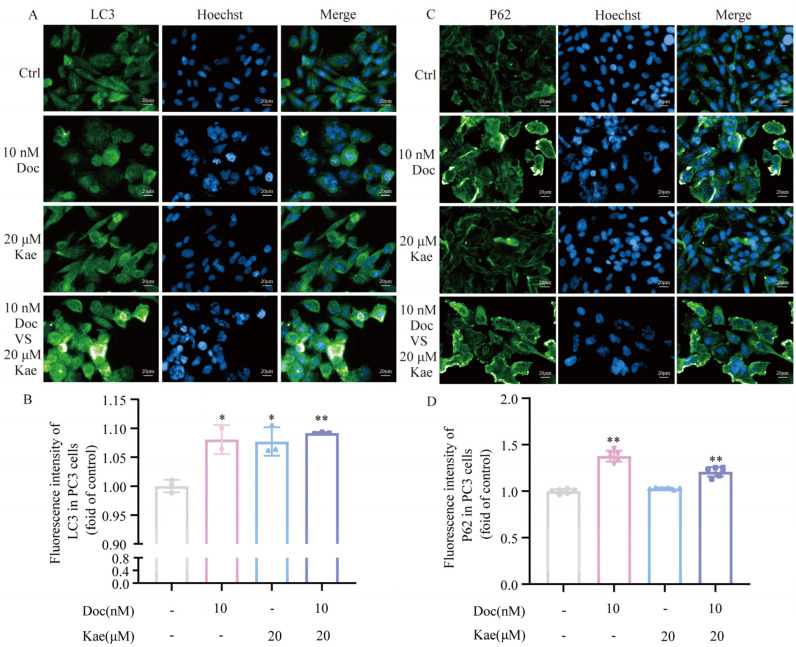
The fluorescence intensity of LC3 protein and p62 protein in PC-3 cells. (**A**,**B**) Effect of different treatments on LC3 protein expression in PC-3 cells. (**C**,**D**) Effect of different treatments on p62 protein expression in PC-3 cells. * *p* < 0.5, ** *p* < 0.01 vs. Ctrl.

**Figure 5 ijms-24-14519-f005:**
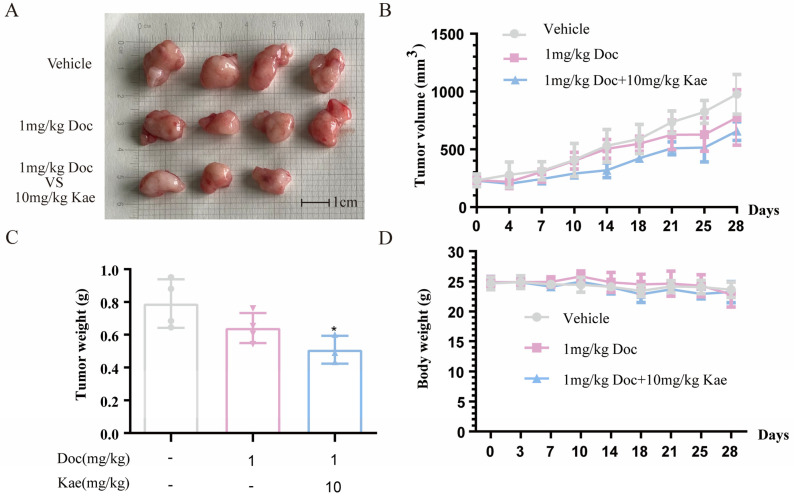
Anti-prostate cancer effect of kaempferol combined with docetaxel in vivo. (**A**) Tumors from nude mice in different treatment groups; (**B**) change in tumor volume in nude mice during administration; (**C**) tumor weight of nude mice at the end of the administration treatment; (**D**) changes in body weight of nude mice during drug administration. * *p* < 0.5 vs Vehicle.

**Figure 6 ijms-24-14519-f006:**
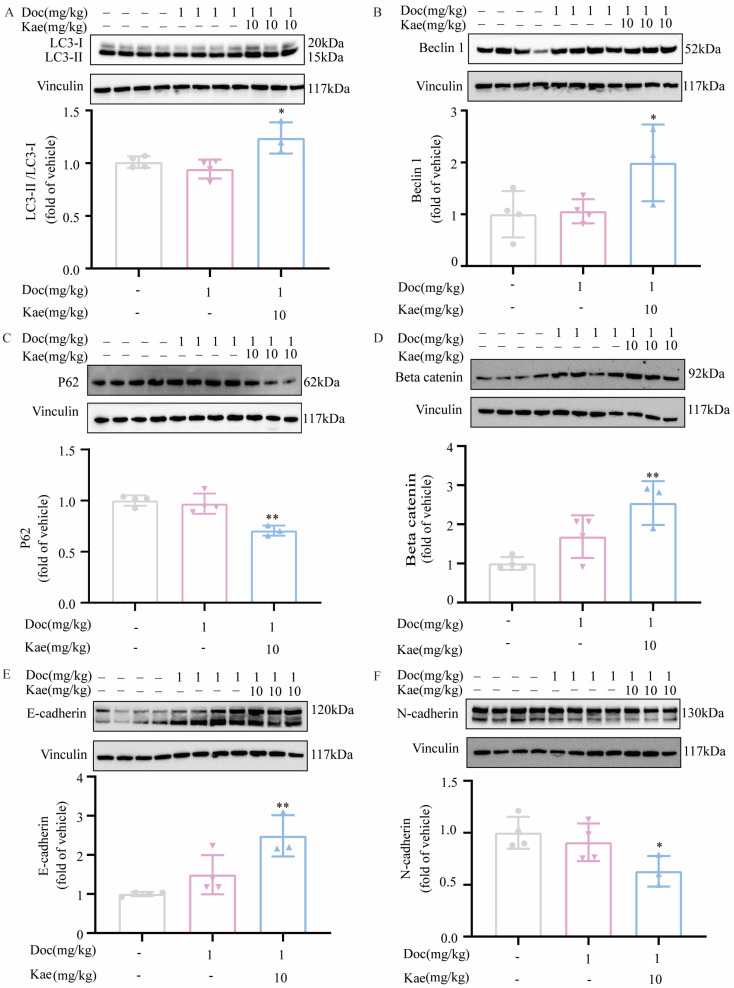
Changes in protein expression in tumor tissue of nude mice after drug administration. (**A**) LC3-II/LC3-I ratio in tumor tissue of nude mice; (**B**) beclin1 expression in tumor tissue of nude mice; (**C**) Beta Catenin expression in tumor tissue of nude mice; (**D**) p62 expression in tumor tissue of nude mice; (**E**) E-cadherin expression in tumor tissue of nude mice; (**F**) N-cadherin expression in tumor tissue of nude mice. * *p* < 0.5, ** *p* < 0.01 vs. Vehicle.

## Data Availability

Uncropped images of the Western blot are available in the Appendix A and other raw data are available on request.

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
