# Peer review of "Combination of Kaempferol and Docetaxel Induces Autophagy in Prostate Cancer Cells In Vitro and In Vivo"

_ijms, 2023, doi:10.3390/ijms241914519_

Round 1

Reviewer 1 Report

Please find attached pdf file of my review report.

Moderate editing of English language is required.

Author Response

Dear reviewer,

Thank you for your comments concerning our manuscript entitled “Combination of kaempferol and docetaxel induces autophagy in prostate cancer cells in vitro and in vivo”. (ID: ijms-2570063). Those comments are all valuable and very helpful for us to revise and improve our paper. We studied comments carefully and have made related corrections in the revised manuscript.

The responses to the comments are listed in the response file.

Reviewer 2 Report

In their paper Zhou et al describe the induction of autophagy in prostate cancer cells both in vitro and in vivo through the use of kaempferol and docetaxel. Although their findings may be of interest, a major revision is required to meet the standards of the journal. Many parts of the paper need to be re-written and some extensive refining of experiments would be necessary to have a publishable version of this work. More specifically, the paper has both minor and major issues.

Minor issues:

-The article discusses the induction of autophagy in prostate cancer cells both in vitro and in vivo through the use of kaempferol and docetaxel. However, it is noteworthy that the introduction of the article lacks any mention of the concept of autophagy. In fact, the autophagic process remains unaddressed throughout the discussion as well. Please, kindly provide a concise introduction of the autophagy after its initial mention to clarify its meaning

-Figure 2: In the part A, quantification of the number of structures (autophagosomes and autolysosomes) present in each condition is required.

-Figure 2: In the part B, kindly review the condition “kae (20 µM)” closely, as an incongruity appears to exist between the Western blot results of LC3 ratio depicted in the image and the subsequent quantification data.

-Figure 3: Quantification of the number of structures (autophagosomes and autolysosomes) present in each condition is required.

-Figure 5: In this figure the authors specify that Western blot experiments were conducted on nude mice. However, the specific organ under analysis remains nuclear.  It is crucial for the authors to explicitly indicate the exact tissue being examined, whether it be the liver, muscle, heart, or another organ. Furthermore, it is necessary to incorporate Western blot analysis of p62, according to results shown in Figure 2.

Major issues:

-To correctly analyze the induction of autophagy in cells, it is necessary to measure the autophagic flux by western blotting in the different treatments shown in the article. To facilitate the understanding of the experiment, see the following article: 10.3390/ijms18091865

- On the one hand, to validate and corroborate the results obtained through western blotting in cells, it would be essential to perform p62 and LC3 immunofluorescence analysis on PC-3 cells in the indicated conditions in figure 2.

-On the other hand, it is important to investigate whether the observed results from western blotting analysis of beclin1 and LC3 protein levels correlate with changes in the expression of these markers at the mRNA level. In this sense, it is necessary to perform quantitative PCR (qPCR) analysis of these markers in the tissue of nude mice (comparing the different treatments). In addition, it is necessary to include also p62.

-In order to enhance the comprehensibility of the results for readers of the IJMS journal, it is necesarry to incorporate a concluding figure that summary the changes demonstrated throughout the article.

Author Response

(The authors gave the same response as above.)

Round 2

Reviewer 1 Report

 After revision the MS is improved and I suggest to the editor to accept the MS.

Minor editing is needed.

Author Response

Dear reviewer,

Thank you for reviewing our manuscript. We appreciate all your comments and suggestions that helped us improve the manuscript. Thank you again for your work.

Reviewer 2 Report

In the revised manuscript, the authors have addressed certain recommendations I previously provided. However, I must express my surprise at the apparent disregard for the majority of my suggestions. I find it perplexing that not a single one of the proposed experiments has been conducted. Given this situation, I strongly emphasize the importance of a comprehensive reevaluation by the authors. It is necessary for them to incorporate the experimental suggestions I have put forth to ensure the attainment of a version suitable for publication in IJMS.

Minor issues:

-The article discusses the induction of autophagy in prostate cancer cells both in vitro and in vivo through the use of kaempferol and docetaxel. However, it is noteworthy that the introduction of the article lacks any mention of the concept of autophagy. In fact, the autophagic process remains unaddressed throughout the discussion as well. Please, kindly provide a concise introduction of the autophagy after its initial mention to clarify its meaning

Response: We have added autophagy-related content to the Discussion section.

-The article continues without introducing the term autophagy in the introduction. Please add it in the second revised version of the manuscript.

-Figure 2: In the part A, quantification of the number of structures (autophagosomes

and autolysosomes) present in each condition is required.

Response: Thank you for your comments. Autophagosomes and autophagic

lysosomes are evidence that autophagy occurs, so we used transmission electron

microscopy to see if they appeared after compound treatment. We considered this

technical tool suitable for qualitative rather than quantitative use, so no quantitative

comparisons were made on the results of this experiment.

-Figure 2, part A: I disagree with the authors' response. Many articles utilize these techniques to quantify autophagosomes/autolysosomes. An illustrative example can be found in this reference DOI: 10.1038/s41418-021-00776-1. Please add it in the second revised version of the manuscript.

-Figure 3: Quantification of the number of structures (autophagosomes and autolysosomes) present in each condition is required.

Response: Thank you for your comment. Thank you for your comments. As with the

response in Figure 2, we are not providing quantitative results in the figure here.

As with the petition in figure 2, please add it in the second revised version of the manuscript.

Major issues:

-To correctly analyze the induction of autophagy in cells, it is necessary to measure the autophagic flux by western blotting in the different treatments shown in the article. To facilitate the understanding of the experiment, see the following article: 10.3390/ijms18091865

Response authors: Thank you very much for your advice. In this article, we focused on the anti-prostate cancer effects of the combination of kaempferol and docetaxel, and

found that they induced autophagy after the combination of the two of them.

However, how the combination induced autophagy and how it interfered with the

apoptosis-inducing effect of docetaxel remains an unanswered question, and these

questions involve complex mechanisms, which we will reveal in the next study.

- In accordance with the title put forth by the authors, "Combination of Kaempferol and Docetaxel Induces Autophagy in Prostate Cancer Cells In Vitro and In Vivo," it is necessary to validate the induction of autophagy in prostate cancer cells resulting from treatments involving kaempferol and docetaxel. It is therefore essential to include the experiment I previously proposed in the earlier version of the review.

- On the one hand, to validate and corroborate the results obtained through western blotting in cells, it would be essential to perform p62 and LC3 immunofluorescence analysis on PC-3 cells in the indicated conditions in figure 2.

Response: Thank you for your comment. Western blot and cellular immunofluorescence are both protein level experiments, and the fluorescence will be quenched during the process of taking pictures, we believe that the quantification of western blot is more accurate than cellular immunofluorescence, so we used western blot for protein detection

- Again I disagree with the authors. It is common practice to corroborate results obtained through one technique by utilizing a distinct method to ensure the validity of findings and to avoid potential experimental artifacts. In this context, numerous scientific publications exist within the literature that reinforce Western blot findings through the application of immunofluorescence techniques. Notably, there are antibodies, such as anti-SQSTM1 (M01, clone 2C11) from Abnova Corporation (Cat# H0000887878-M01) or anti-LC3 from Novus (Cat# NB600-1384), which demonstrate reliable performance for conducting these experiments, thereby mitigating concerns regarding fluorescence quenching issues. Please add it in the second revised version of the manuscript.

Author Response

(The authors gave the same response as above.)

Round 3

Reviewer 2 Report

Considering the modifications made by the authors, I believe that the manuscript is now ready to be published.